# Vapour Pressure Deficit (VPD) Drives the Balance of Hydraulic-Related Anatomical Traits in Lettuce Leaves

**DOI:** 10.3390/plants11182369

**Published:** 2022-09-11

**Authors:** Chiara Amitrano, Youssef Rouphael, Stefania De Pascale, Veronica De Micco

**Affiliations:** Department of Agricultural Sciences, University of Naples Federico II, 80055 Naples, Italy

**Keywords:** leaf anatomical traits, leaf hydraulic conductance, stomatal density, vein density, VPD

## Abstract

The coordination of leaf hydraulic-related traits with leaf size is influenced by environmental conditions and especially by VPD. Water and gas flows are guided by leaf anatomical and physiological traits, whose plasticity is crucial for plants to face environmental changes. Only a few studies have analysed how variations in VPD levels influence stomatal and vein development and their correlation with leaf size, reporting contrasting results. Thus, we applied microscopy techniques to evaluate the effect of low and high VPDs on the development of stomata and veins, also analysing leaf functional traits. We hypothesized that leaves under high VPD with a modified balance between veins and stomata face higher transpiration. We also explored the variability of stomata and vein density across the leaf lamina. From the results, it was evident that under both VPDs, plants maintained a coordinated development of stomata and veins, with a higher density at low VPD. Moreover, more stomata but fewer veins developed in the parts of the lettuce head exposed to light, suggesting that their differentiation during leaf expansion is strictly dependent on the microclimatic conditions. Knowing the plasticity of hydraulic-related morpho-functional traits and its intra-leaf variability is timely for their impact on water and gas fluxes, thus helping to evaluate the impact of environmental-driven anatomical variations on productivity of natural ecosystems and crops, in a climate change scenario.

## 1. Introduction

Plant traits are the morpho-anatomical, physiological, phenological and biochemical characteristics that can be measured at the single-organism level [1]. These traits depend on the genetic properties of the species, reflecting their evolutionary lineage, and are deeply influenced by the environment [2,3]. Plant traits have gained popularity because of their use as proxies in vegetation modelling to predict plants performance under changing climate conditions [4,5]. Among plant traits, leaf size has a central role in plant acclimation to environmental conditions [6,7,8]. Variation in leaf size has been found along climatic gradients, often with increments in lamina expansion in humid habitats [9,10]. Alternatively, small leaves are more easily found in dry environments, since they can react to high irradiance reducing transpiration costs [11]. Indeed, the advantage of having smaller leaves is correlated to the thickness of the boundary layer, which increases with leaf size, so it would be difficult for bigger leaves to reduce the heat loads in a dry environment. Bigger leaves in a dry environment will face more serious risk of overheating, due to excessive light energy and little available water [12]. Therefore, irradiance during plant growth and the evaporative demand (expressed as the air vapour pressure deficit; VPD) that the plants are subjected to are strictly connected, affecting not only the whole plant–water relationships, but also the development of leaf size and other plant anatomical traits [13]. Indeed, leaves under high light may suffer from risks caused by high VPD and dehydration, and the plasticity of their leaf water-related traits can contribute to maintain an efficient photosynthesis under limiting environmental conditions [12]. Furthermore, developing small leaves in a dry environment optimizes the whole plant resources allocation. For a plant, developing narrower leaves allows saving costs attributed to cell-walls and to a wide vascular system, giving a competitive advantage in hostile environments where the access to carbon and water resources is already inadequate [9,14]. Furthermore, over the years, leaf size has been positively or negatively correlated with other leaf morpho-anatomical traits linked to plant hydraulics, such as: stomatal and vein densities, often with controversial results mainly depending on the species and cultivar studied (also due to different strategies associated with the specific photosynthetic pathway). Just to mention a few studies, Gupta [15] was among the first to show that in five Solanaceae plants, the average number of stomata per unit area as well as vein density were negatively correlated to the leaf area [16]. Carins Murphy et al. [14] proposed that in species with high leaf plasticity, which promptly adapts to changes in environmental conditions, stomatal and vein density should be “diluted” during leaf expansion, thus allowing coordinated changes in vein and stomatal densities. These coordinated changes would allow the maintenance of a high physiological efficiency (photosynthesis, conductance, water use efficiency) of the species under sub-optimal/harsh environmental conditions [17]. However, with regard to the relations among leaf size, vein and stomatal density showed variations at high and low VPDs. Carins Murphy et al. [18] found that leaf anatomical traits of *Toona ciliata* were independent from leaf size under high and low VPD, while other research evidenced an enhanced stomatal density, but reduced vein density in *Rosa hybrida* under high VPD [19] or reduced stomatal and vein densities coordinated with a reduced leaf area in *Vigna radiata* under high VPD [20]. Differences were even found in different cultivars of *Solanum lycopersicum* with reduced stomatal density and leaf area and no differences in vein density in ‘Jinpeg’, and reduced stomatal density, enhanced vein density and no differences in leaf area in ‘Zhongza’ [21].

Generally, besides some intra- and inter-species variation, leaf morpho-anatomical traits are different in sun and shade leaves, usually with the following general pattern, under favourable environmental conditions: leaves developed under sun have less expanded but thicker lamina, higher stomatal density, and a well-structured palisade tissue, often showing thin grana stacks in their chloroplasts, compared to shade leaves [22,23]. Although vein density has been much less studied so far than other traits, there is evidence that sun leaves evolve higher vein density than shade leaves [14,17,24]. In addition to morpho-anatomical traits, sun and shade leaves also differ in several physiological traits. Sun leaves usually present a higher saturation point of photosynthesis and chlorophyll a/b ratio [25,26]; whereas shade leaves usually have higher photoprotection capacity based on non-photochemical quenching (NPQ) and a higher amount of chlorophyll molecules per leaf dry mass [27].

Moreover, the hypothesis that vein and stomatal densities are related to both leaf size and irradiance have been long debated in the past, since plasticity in leaf size has influence on its hydraulic-related traits and could provide a way for plants to contrast the different evaporative demand of sun and shade [14]. However, the regulation of this phenotypic plasticity during plant development remains still unclear [28]. For instance, Hoshino et al. [28], studying *Arabidopsis* leaf thickening during sun-leaf development, proposed that for the differentiation between sun and shade leaves, the early stages of development are crucial, since anisotropic cell enlargement occurs only in sun leaves in the first 5–7 days of development, primed by high light intensity. Shade leaves instead skip anisotropic cell elongation and undergo isotropic growth in a late phase of leaf thickening. To the best of our knowledge, so far, no studies have explored the development of these traits in the same leaf, especially in crop species.

All these traits, however, have been poorly explored in crops, where only a few studies can be found, mostly in greenhouse trials of tomatoes [21,29], and little is known about lettuce morpho-anatomical development under different environmental conditions. Moreover, recently many models have been developed to forecast crop behaviour in greenhouses due to changes in the environmental conditions, and to be used in decision support systems [30]. Most of these models, however, are based on environmental parameters and do not take into consideration the different development of crop anatomical traits under different environments [31]. The importance of considering plant anatomical traits for precise understanding of plant physiological acclimation is more and more recognized and has recently led to the development of “Anatomics” as a new phenotyping strategy based on the quantification of plant anatomical traits, with promising applications not only in fundamental plant biology, but also in agriculture and ecology [32].

In this study, we compare the leaf plasticity in size, density of stomata and veins of 2 butterhead lettuces with green and red leaves (*Lactuca sativa* L. var. *capitata*) grown under different VPDs (low and high). After developing the first two leaves, lettuce plants continue to form the rosette pattern that it grows in, creating in the same leaf shade and sun areas subjected to a different boundary layer, relative humidity and, overall, to different microclimatic conditions, until developing into a compact head. Thus, we hypothesized that if humidity (low VPD) was the main factor in triggering stomatal and vein density, the highest density should be found in the shaded parts of the leaf, vice versa if light was the main factor, the highest density should be found in the light-exposed parts. In the light of the above, the main questions of this study were: (i) Are vein and stomatal densities diluted with leaf size in lettuce? (ii) Do the different microclimatic conditions around the leaf (i.e., VPD and light) prime the development of different patterns of morpho-anatomical traits along the lamina? Moreover, this study will help in gaining insight and data to be further applied into a developmental model which will consider the cultivar-specific characteristic of the species under different environments as an input parameter. Assessing the impact of different air humidity and irradiances on leaf anatomical characteristics and hydraulic-related traits will be an important starting point to evaluate to what extent the intra-leaf anatomical spatial variability would influence the key regulator role of such traits in fundamental physiological processes and ultimately the species capacity to acclimate to the ongoing climate change conditions.

## 2. Results

### 2.1. VPD Influence on Leaf Size and Functional Traits

VPD had a significant effect on leaf size (LS), leaf mass per area (LMA), leaf dry matter content (LDMC) and relative water content (RWC) (*p* < 0.001) (Table 1). Alternatively, the cultivar and the interaction between main factors (VPD × C) had a significant effect only on RWC (*p* < 0.05, *p* < 0.01, respectively) (Table 1). More specifically, leaves from plants grown under low VPD (LV) were 22% more expanded than those grown under high VPD (HV) (Table 1). These leaves also presented increased LMA, LDMC and RWC (51%, 53% and 59% more than HV, respectively). Concerning RWC, significant differences were also found between the G and R cultivars, with increments in R by 9%. No significant differences were detected among treatments concerning equivalent water thickness (EWT).

### 2.2. VPD Influence on Leaf Stomatal and Vein Traits

VPD had a significant effect on stomata and vein traits influencing stomatal length (SL), epidermal cell density (ED) and free vein ending (FEV) (*p* < 0.001) (Table 2). Differently, the cultivar and the interaction between main factors (VPD × C) had a significant effect on SL and FEV (*p* < 0.05). LV also elicited the development of more FEV and ED (10 and 50% more than HV), but smaller stomata (11% reduction in length of guard cells compared to HV) (Table 2). Concerning SL and FEV, significant differences were also found between the green (G) and red (R) cultivars, with higher values in the G cultivar by 6, 27 and 24% compared to R. No significant differences were detected among treatments concerning stomatal width (SW). Moreover, VPD had a significant effect on stomata and veins per leaf area (SD × LS and VLA × LS) (*p* < 0.01); the cultivar and the interaction between factors (VPD × C) had a significant effect on SD × LS (*p* < 0.05) and on VLA × LS (*p* < 0.01) (Appendix A). In general, LV plants presented more stomata and veins per leaf size with values 38 and 36% higher than HV, respectively (Figure 1a,b).

### 2.3. Relationships between Leaf Morpho-Anatomical Traits

As presented in Figure 2a,b, both vein and stomatal densities per leaf size (VLA × LS and SD × LS) values plotted against leaf size deviate from the proportional relationships (broken line). Only the epidermal cells density per leaf size (ED × LS) maintained a proportional relationship with leaf size in all the conditions (LVG, LVR, HVG, HVR) (Figure 2c). Moreover, there was a strong positive relationship between vein and stomatal density in all the conditions (Figure 3). At low VPD, a significant correlation between vein and stomata in both G and R plants (R^2^ = 0.96, and 0.88, *p* < 0.001) was found, which was based on lower values of both parameters compared to high VPD where the relation was highly significant too (R^2^ = 0.87 and 0.93, *p* < 0.001) (Figure 3).

### 2.4. VPD Influence on Stomatal and Vein Distributions within Leaves

The distribution of veins and stomata within the lamina changed with the VPD, the cultivar and the position within the leaf (Table 3 and Table 4). More specifically, VPD and position as main factors influenced SD, SL, FEV and VLA (*p* < 0.001). Differently, the cultivar only influenced the vein traits (*p* < 0.01). Concerning stomatal density (SD), highest values were found at the apex part of low VPD plants in both green and red cultivars (LVGa and LVRa), followed by the apex part of high VPD green and red plants (HVGa and HVRa) and then by the medium part of low VPD green and red plants (LVGm and LVRm), which in turn were higher than the medium part of high VPD green and red plants (HVGm and HVRm). The lowest values were found in high VPD green and red plants at the bottom part (HVGb and HVRb). Alternatively, stomatal length (SL) showed an opposite trend, being higher under high VPD of both green and red plants at the bottom part (HVGb and HVRb), followed by high VPD green and red plant at the medium (HVGm, HVRm) and at the apex (HVGa and HVRa) parts (with no differences among them). The lowest values were found in low VPD of green and red plants at the apex part (LVGa and LVRa).

Free vein endings (FVE) showed higher values at low VPD of green and red plants at the apex part (LVGa and LVRa)followed by low VPD of green and red plants at the medium part (LVGm and LVRm) and by low VPD of green and red plants at the bottom part (LVGb and LVRb). The lowest values were found in high VPD of green and red plants at the bottom part (HVGb and HVRb). Alternatively, vein density (VLA) was enhanced in the bottom part of low VPD green and red leaves (LVGb and LVRb), which in turn were higher than low VPD of green and red leaves at the medium (LVGm and LVRm) and apex (LVGa and LVRa) parts. The lowest values were found in high VPD of green and red plants at the apex part (HVGa and HVRa).

Moreover, the vein and stomatal densities of the three leaf portions displayed positive significant relationships (Figure 4 and Appendix A). However, the degree of these relationships varied among treatments (Figure 4 and Appendix A). Increasing the VPD induced a decrease in vein and stomatal density in HVGb (R^2^ = 0.66), HVRb (R^2^ = 0.92), HVGm (R^2^ = 0.89), HVGm (R^2^ = 0.83), HVGa (R^2^ = 0.94), and HVRa (R^2^ = 0.30), whereas, under low VPD, stomatal and vein density increased and were still strongly correlated: in LVGb (R^2^ = 0.87) LVRb (R^2^ = 0.86), LVGm (R^2^ = 0.85), LVGm (R^2^ = 0.92), LVGa (R^2^ = 0.92), and LVRa (R^2^ = 0.75).

Moreover, moving from b to a, leaves presented reduced vein density and increased stomatal density both under LV and HV (Table 4 and Appendix A). Green and red lettuces showed similar relations in the three portions of the leaf (Appendix A).

## 3. Discussion

### 3.1. Response of Stomatal and Vein Densities and Coordination with Leaf Size under Different VPD

Thus far, only a few studies have analysed how variations in VPD influence stomatal and vein development in the same species, often reporting contrasting results [17,20]. Moreover, very little information can be found in crop species. More specifically, little is known about the direction of response or coordination (positive/negative) among stomatal and vein densities and their correlation with leaf size. Here, we found increased leaf size under LV compared to HV with no differences between cultivars (Figure 1), clearly indicating that leaf size is strictly dependent on VPD. LV leaves also presented increased values of RWC, probably indicating a better water availability and use. Usually, VPD, rainfall and temperature influence leaf size, and the global trend is to develop smaller leaves in drier environments [5]. Under dry air (high VPD), plants with smaller leaves, and a thinner boundary layer easily reduce their heat loads and water demand [33]. Moreover, under a dry environment, the relative water content of leaves is commonly reduced [34,35]. According to our findings, other greenhouse and indoor trials have also reported smaller leaf size in crops subjected to high VPD levels compared to low VPDs [36,37]. Our results are in agreement with recent research [14,18], showing no proportional coordination among stomatal density and leaf size as well as vein density and leaf size (Figure 2). In the last decades, other studies have supported this idea; for instance, Scoffoni et al. [38] found an independence of minor vein density from leaf size in 10 different species of moist and dry habits. This was confirmed by Sack and Scoffoni [39] in a study on more than 100 dicotyledonous species. Moreover, here we found a coordination between vein and a stomatal density under both VPD conditions, and more specifically an increased density of both anatomical traits in plants under low VPD, the same plants which presented a higher leaf size. In an under optimal environmental condition, an adequate balance between stomata and vein should be necessary since leaf venation must be enough to supply water to stomata and replenish the water loss due to transpiration in order to maintain an adequate physiological function [17,38,40]. Most likely, the coordinated development of vein and stomatal densities under both VPDs is an adaptation mechanism of the species trying to maintain the water balance under favourable (LV) and less favourable (HV) environments.

Moreover, in our study, there were no significant differences in epidermal cell size between treatments; however, epidermal cell density per leaf area increased under low VPD and was positively correlated with leaf size both under LV and HV (Figure 2c), meaning that epidermal cell number per unit surface is proportional to changes in leaf area. This suggests that VPD plays a role in regulating cell cycles during leaf lamina expansion, ultimately resulting in a control of cell number more than of cell size. Furthermore, we observed higher stomatal and vein densities at low VPD, suggesting that the plant regulates the construction of vein and stomata under different VPD conditions by actually controlling the number of differentiating cells. The higher stomatal and vein densities at low VPD (higher leaf size; Figure 1 and Appendix A, Table 3) are also another sign supporting that these anatomical traits are not diluted with VPD-induced increasing leaf size. Moreover, in our study, there was a reduction in stomatal length under low VPD (Table 2), also in agreement with the trend of decreasing epidermal cell size occurring in the same conditions. Similarly to our results, smaller stomata have been previously associated with high densities [41]. However, Giday et al. [42] found that stomatal size and density are not correlated in rose. Size, other than density, is an important trait, since it influences the opening/closing reaction of stomata in response to environmental conditions [43]. When stomata increment their volume, the surface area to volume ratio decrease and this has been listed as the main reason for their slower response [44]. However, smaller stomata usually show shorter response times and can optimize water fluxes under limiting water availability [37]. In our study, other measured leaf traits (LMA, LDMC) were enhanced in LV (Table 1). These traits vary strongly with light, temperature, CO_2_ concentration and water availability [45]. In particular, LMA depends on both LDMC and leaf thickness so much that dividing LMA by LDMC often provides a good estimation of leaf thickness [46]. Moreover, changes in LDMC have been related to water availability. Therefore, a correlation with other traits (stomatal and vein densities and leaf conductance) has been found [47]. The same correlation happens here with higher LDMC associated to higher stomatal and vein density in LV plants. The coordination between leaf hydraulic-related traits (stomatal density and size, vein density) and the coordination between these anatomical traits and other leaf traits under different VPDs is fundamental because it represents a clear indication of how environmental conditions play a role in the adaptation of leaf anatomical traits.

### 3.2. Acclimation of Anatomy to Sun and Shade within the Same Leaf

In this study, we have examined hydraulic-related traits of lettuce plants throughout the whole leaf lamina, from the bottom to the apex. Lettuces grow as a rosette, creating in the same leaf different microclimatic conditions, with only the apex part exposed to light. It is known that sun and shade leaves do balance different anatomical traits to acclimate to high and low irradiance. Shade leaves are subjected to lower evaporative demand and are in need to maximize light capture, absorption, and processing [23]. From a morpho-anatomical point of view, leaves developed under shade are usually thinner, with lower stomatal and vein densities than sun leaves [22,48]. Our results showed that under different VPDs, lettuce plants encountered a different coordination between stomatal and vein densities along the leaf lamina. While stomata increased in the part exposed to light (from bottom to apex) independently of the VPD, veins were always denser in plants exposed to low VPD along the whole leaf (increasing from the apex to the bottom) (Table 3). This likely indicates that light has a stronger influence on stomatal development, while relative humidity would have the major influence on vein development. Indeed, in both VPD conditions, stomatal density incremented towards the apex (light exposed) of the leaf lamina, whereas veins incremented in the bottom part and were always denser at low VPD (higher relative humidity). Indeed, in the bottom and middle parts of the leaf, covered by other leaves, the microclimatic condition is characterized by a higher relative humidity (Low VPD) and a thicker boundary layer. Different positions in the canopy have proved to change anatomical and hydraulic-related traits in trees [49], along a gradient which affects the whole plant photosynthetic capacity and yield [50]. In agreement with our results, a high density of stomata has been found in sun leaves of different species (tomatoes, sorghum, coffee) [22,51,52]; however, very little is known about vein development within leaves and environmental conditions.

The combined influence of light and VPD should be explored further since it could be responsible for the different degree of coordination between these traits. The adjustment of these hydraulic-related traits with microclimatic conditions is fundamental to provide optimal water and gas fluxes throughout the entire plant [49,53]. The overall analysis on lettuce evidenced the occurrence of a significant intra-lamina variability of the leaf traits coordination due to the microclimatic conditions which therefore influence eco-physiological plant behaviour [54]. These findings strengthen the recent claim that *Anatomics* needs to be further developed and applied to parameterize functional–structural models simulating the impact of the variation in anatomical traits on plant growth and physiological processes [32]. This is timely and valuable, especially to evaluate the impact of environmental-driven anatomical variations on productivity of natural ecosystems (e.g., in semi-arid regions), and crops, in the climate change scenario we are facing today.

## 4. Materials and Methods

### 4.1. Plant Material and Growth under Controlled Conditions

The study was conducted on 2 varieties of butterhead Salanova^®^ lettuces (*Lactuca sativa* L. var. *capitata*), one with green and the other with red leaves. Lettuce was chosen because it is the most common leafy green crop cultivated in a controlled environment and, to our knowledge, no studies have investigated the development of leaf anatomical traits in relation to micro-environmental conditions. Seeds were provided by Rijk Zwaan (Rijk Zwaan, Westland,, The Netherland) and showed 100% germination. The experiment was carried out in a growth chamber (KBP-6395F, Termaks, Bergen, Norwey) under a photoperiod of 12h. Light was provided by and RGB LED panel with an intensity of 315 PPFD µmol m^−2^ s ^−1^ at the canopy level. Eighteen lettuces (9 green and 9 red) were grown for 23 days in two different trials under the same temperature (T) of 24 °C, but different air relative humidity (RH), resulting in two different VPDs. In the first trial, an average VPD of 0.69 kPa (Low VPD) was kept, while the second trial was conducted at an average VPD of 1.76 kPa (High VPD). Temperature and humidity were monitored throughout the whole experiment duration by using sensors equipped with a data logger (Testo 174H). Daily rotation of the trays ensured homogenous light and humidity conditions across the shelf surface. Plants were watered daily to field capacity. All the analyses were carried out on 10 fully expanded leaves per VPD condition. A list of all the measured traits and their unit of measurements are reported in Table 5.

### 4.2. Leaf Size and Other Leaf Functional Traits

Leaf functional traits were evaluated following Cornelissen et al. [55]. Firstly, leaves were scanned to calculate leaf size (LS; lamina area in mm^2^) using ImageJ software (national Institutes of Health, Mongomery, MD, USA). Then, the fresh weight (FW) of each leaf was recorded and the leaf petiole was submerged in distilled water in the dark for 48 h and then re-weighted to calculate the saturation weight (SW); the dry weight (DW) was obtained by oven-drying leaves at 60 °C for 72 h, until they reached a constant weight. These parameters were used to evaluate: the water status of the leaves, measured as relative water content (RWC %) and expressed as percentage of (FW−DW)/(SW−DW); the leaf dry matter content (LDMC) considered a proxy for leaf tissue density [56] and expressed as (DW/SW) in gg^−1^; leaf mass per area (LMA), calculated as the ratio between DW and LS (g mm^−2^), which is used as a proxy for sclerophylly [57]; and the hypothetical thickness of a single layer of H_2_O on the leaf area, equivalent water thickness (EWT), calculated as (FW−DW)/LS and expressed in mg mm^−2^. As water is involved in all physiological processes, EWT reflects the physiological status of vegetation and is also correlated to other key water status parameters, including the canopy water content (CWC), which is used for estimating the effects of climate change [58].

### 4.3. Leaf Vein Traits

To analyse morpho-anatomical variability throughout the leaf lamina, each leaf was collected and then divided into three parts (one third per part, along the main axis) called bottom, b; middle, m; and apex, a (Figure 5). All the sampling for anatomical analyses were performed at the same time, at the end of the cultivation cycle, during light hours, taken quickly to standardize collection across all conditions. To determine vein traits, the entire leaves were chemically cleared with 5% NaOH in aqueous solution and bleached in EtOH dilution series, following Miksche and Berlyn [59]. In order to highlight even the smallest veins, cleared leaves were submerged in 1% safranin in EtOH for 10 min and gently rinsed with 100% EtOH before being stained in 1% fast green in ETOH for a few seconds and rinsed again with 100% EtOH. By means of a transmitted light microscope (BX51; Olympus) equipped with a camera (EP50; Olympus), bottom, middle and apex of each leaf was imaged in 5 fields of view at a magnification of 4x (image area 24 mm^2^). From those pictures, vein density (VLA, total vein length per mm^2^ of leaf area) and free vein endings per area (FEV) were calculated using image J software, following Sack and Scoffoni [11]. In brief, VLA was calculated as the ratio of the sum of vein lengths of 4th order veins and higher and the difference between the area of the image and the area occupied by the 2nd order veins, expressed in mm mm^2^; FEV was quantified as the ratio of the number of free vein endings and the difference between the area of the image and the area occupied by the 2nd order veins, expressed in n mm^2^. These data are shown in Table 2 and Table 3, Figure 4, Appendix A. Moreover, the vein length per leaf size (VLA × LS) was also calculated and expressed in mm mm^−2^ LS. These data are show in Figure 1, Figure 2 and Figure 3 and Appendix A.

### 4.4. Stomatal Traits

Stomatal traits were determined on leaf abaxial peels, taken centrally in each region of the leaf avoiding the midrib as well as the margin. For each leaf, measurements were averaged from 5 regions obtained from 3 different peels of the apex, middle and bottom (a, m, b) portion of each leaf. Each field view was set at 20× magnification (field area of 0.95 mm^2^). Stomatal density (SD) was quantified as number of stomata per unit leaf area and expressed in mm^2^ using ImageJ. Stomatal guard cells length (SL) and width (SW) were determined considering the guard cell length (pole-to-pole) (µm), of 5 stomata per field, at a magnification of 40× (filed area of 0.15 mm^2^). Moreover, the density of epidermal cells (ED) was quantified at 20× magnification and expressed as n mm^2^. All the measurements were performed in three fields per 5 leaf samples, being careful to avoid main veins or tissue defects. These data are shown in Table 2 and Table 3 and Figure 4, Appendix A.

As for veins, stomatal and epidermal cell number per leaf size (SD × LS and ED × LS) were quantified by multiplying the density of stomata, and epidermal cells by leaf size as reported in Carins Murphy et al. [18]. These data are reported in Figure 1, Figure 2 and Figure 3 and Appendix A.

### 4.5. Relationships between Leaf Size and Leaf Hydraulic-Related Traits

The coordination between leaf size and leaf hydraulic-related traits (stomatal and vein density per leaf size) was assessed by plotting the stomatal density against 1/leaf size and vein density against 1/√leaf size as reported in Carins Murphy et al. [18] (i.e., quantified as the deviation from a proportional deviation). The coordination between the epidermal cell density per leaf and leaf size was tested for proportionality in the same way. These data are reported in Figure 2.

### 4.6. Statistical Analysis

Leaf morpho-anatomical traits data of green and red Salanova plants grown at low and high VPD were analysed by a two-way analysis of variance (ANOVA) considering the VPD and the cultivar as main factors. A three-way ANOVA was then performed on stomata and vein traits reported in Table 3 and Table 4, considering VPD, cultivar and position within the leaf (a, apex; m, medium; b, bottom) as main factors. The Kolmogorov–Smirnov and Shapiro–Wilk tests were performed to check for normality and a Levene’s test of homogeneity was used to determine if samples had equal variance. Tukey post hoc test was used for means separation (*p* < 0.05). All statistical analyses were performed with SPSS 13 statistical package (SPSS, Chicago, IL, USA). Furthermore, correlations between vein and stomatal density were also calculated: firstly, the correlations between vein density and stomatal density were reported for green and red plants grown under low and high VPD. Secondly, within the single leaf, correlations among vein and stomatal densities were also calculated in the three different positions (a, m, b). Pearson rank correlation coefficient was calculated.

## 5. Conclusions

The results of this study suggested that the VPD triggers a different response in lettuce plants in terms of leaf traits’ development. Vein and stomatal densities showed a coordinated response under high and low VPD with increments in veins and stomatal densities with decreasing VPD. Furthermore, we found positive relationships between vein density and leaf size as well as between stomatal density and leaf size, confirming our hypothesis that, in lettuce plants, both stomatal and vein densities are not diluted with leaf size. In the present study, the second hypothesis was also confirmed, since irradiance had a predominant role (compared to VPD) in triggering the formation of stomata, but not of veins. Indeed, in the apex part of the leaf, the only one exposed to light, stomatal density was the highest despite the differences in VPD. However, the same relationship does not apply to vein density, which was higher under low VPD, especially in the bottom part of the leaves, while maintaining the same pattern of stomata density (i.e., increments in the apex followed by middle and bottom part). Nevertheless, positive relationships were always found in vein and stomatal densities from different parts of the leaf. Consequently, the allocation of veins and stomata during leaf development seems to be strictly dependent on the microclimatic conditions. Further research is therefore needed to understand the developmental basis for these anatomical traits and their coordination in crops. The plasticity in hydraulic-related traits with microclimatic conditions plays a critical role to provide optimal water and gas fluxes and helping plants adapt to changes in the environment.

## Figures and Tables

**Figure 1 plants-11-02369-f001:**
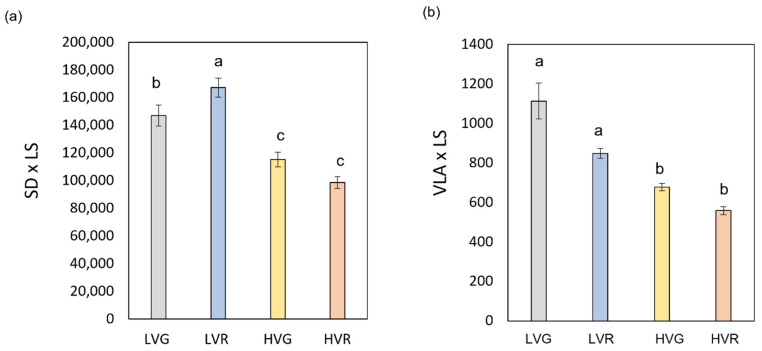
(**a**) Stomatal density per leaf size (SD × LS) and (**b**) vein density per leaf size (VLA × LS) of green and red Salanova lettuces grown under low and high VPD. Mean values ± standard errors are shown. Different letters correspond to statistically significant differences according to Tukey test (*p* < 0.05).

**Figure 2 plants-11-02369-f002:**
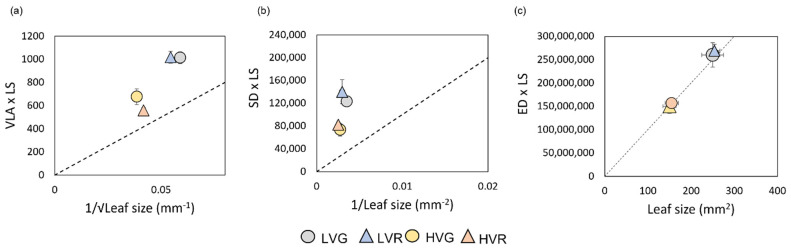
(**a**) Vein density per leaf size (VLA × LS) and 1√leaf size, (**b**) stomatal density per leaf size (SD × LS) and 1/leaf size and (**c**) total epidermal cell number per leaf size (ED × LS) and leaf size of grey (circles) and red (triangles) Salanova lettuces grown under low and high VPDs. Mean values ± standard errors are shown. Broken line represents the proportional relationships.

**Figure 3 plants-11-02369-f003:**
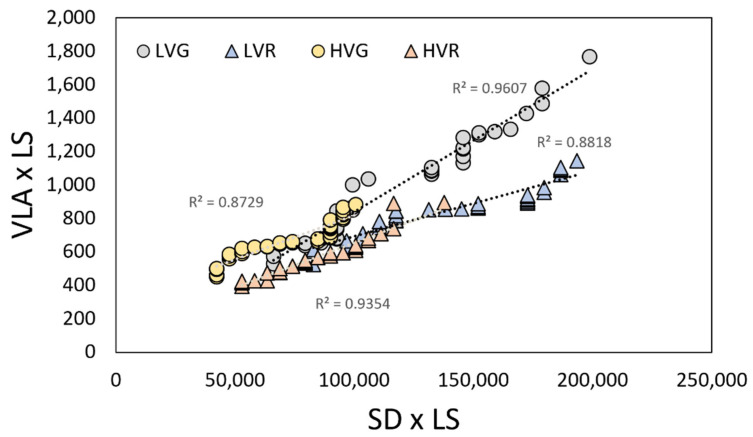
Vein density per leaf size (VLA × LS) and stomatal density per leaf size (SD × LS) of grey (circles) and red (triangles) lettuces grown under low and high VPDs. Regression lines and R2 values are also shown.

**Figure 4 plants-11-02369-f004:**
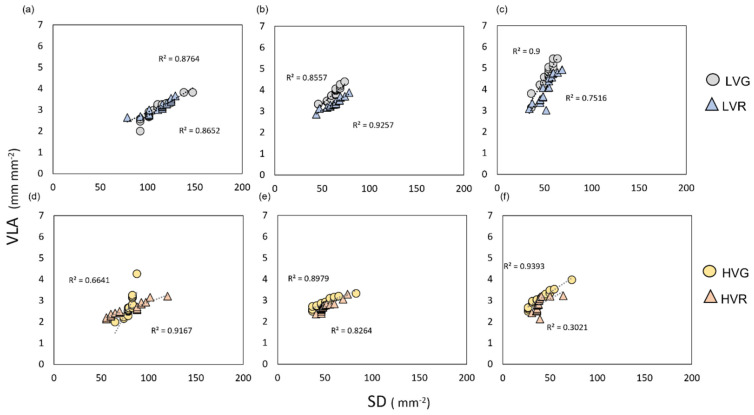
(**a**–**c**) Vein and stomatal density relationships of grey (circles) and red (triangles) lettuce grown under low VPD in the bottom, LVb (**a**); middle, LVm (**b**); and apex, LVa (**c**) part of the leaf. (**d**–**f**) Vein and stomatal density relationships of grey (circles) and red (triangles) lettuce grown under high VPD in the bottom, HVb (**d**); middle, HVm (**e**); and apex, HVa (**f**) part of the leaf. Regression lines and R^2^ values are also shown.

**Figure 5 plants-11-02369-f005:**
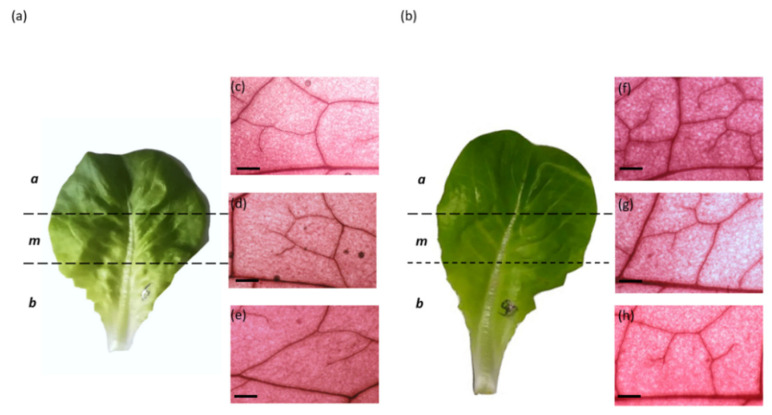
Schematic representation of leaf visual division into three portions: bottom (**e**,**h**), middle (**d**,**g**) and apex (**c**,**f**), from a representative plant grown under (**a**) high and (**b**) low VPD. Representative micrographs of leaf veins for both conditions are shown. Scale bars: 500 µm.

**Table 1 plants-11-02369-t001:** Leaf traits in terms of leaf size (LS), leaf mass per area (LMA), leaf dry matter content (LDMC), relative water content (RWC), equivalent water thickness (EWT) of green (G) and red (R) Salanova plants grown under low (LV) and high (HV) VPD, with no differences within the leaf parts. All data are reported as mean ± se (standard error). Different letters correspond to significant differences according to Tukey test (*p* < 0.05).

	LS (mm^2^)	LMA (mg mm^−2^)	LDMC (g g^−1^)	RWC (%)	EWT (mg mm^−2^)
VPD (V)					
LV	316.8 ± 7.7 ^a^	0.51 ± 0.03 ^a^	0.13 ± 0.008 ^a^	95.1 ± 1.4 ^a^	3.06 ± 0.13 ^a^
HV	247.5 ± 2.5 ^b^	0.25 ± 0.01 ^b^	0.06 ± 0.002 ^b^	87.7 ± 3.4 ^b^	3.55 ± 0.09 a
Cultivar (C)					
G	279.0 ± 6.1 ^a^	0.38 ± 0.02 ^a^	0.09 ± 0.006 ^a^	90.1 ± 2.6 ^b^	3.26 ± 0.10 ^a^
R	285.2 ± 4.1 ^a^	0.37 ± 0.02 ^a^	0.09 ± 0.006 ^a^	92.9 ± 2.7 ^a^	3. 35 ± 0.12 ^a^
Interaction					
LVG	310.1 ± 8.9 ^a^	0.52 ± 0.03 ^a^	0.13 ± 0.007 ^a^	94.3 ± 1.6 ^b^	3.15 ± 0.13 ^a^
LVR	323.5 ± 6.6 ^a^	0.51 ± 0.03 ^a^	0.13 ± 0.009 ^a^	95.1 ± 1.1 ^a^	2.98 ± 0.12 ^a^
HVG	247.9 ± 3.3 ^b^	0.25 ± 0.01 ^b^	0.06 ± 0.002 ^b^	89.2 ± 5.1 ^c^	3.38 ± 0.07 ^a^
HVR	247.0 ± 1.7 ^b^	0.25 ± 0.01 ^b^	0.06 ± 0.002 ^b^	86.2 ± 4.8 ^c^	3.72 ± 0.11 ^a^
Significance					
VPD	***	***	***	***	NS
C	NS	NS	NS	*	NS
VPD × C	NS	NS	NS	**	NS

NS, *, ** and *** Not significant or significant at *p* < 0.05, 0.01 and 0.001, respectively.

**Table 2 plants-11-02369-t002:** Stomata and vein traits in terms of stomatal length (SL), stomatal width (SW), epidermal cell density (ED) and free vein endings (FEV) of green (G) and red (R) Salanova plants grown under low (LV) and high (HV) VPD, with no differences within the leaf parts. All data are reported as mean ± se of 9 replicates. Different letters correspond to statistically significant differences according to Tukey test (*p* < 0.05).

	SL (µm)	SW (µm)	ED (n mm^−2^)	FEV (n mm^−2^)
VPD (V)				
LV	20.28 ± 0.63 ^b^	15.91 ± 0.40 ^a^	1.93 × 10^8^ ± 9.84 × 10^6^ ^a^	5933 ± 46 ^a^
HV	22.55 ± 0.59 ^a^	15.93 ± 0.17 ^a^	1.71 × 10^8^ ± 5.15 × 10^6^ ^b^	2913 ± 16 ^b^
Cultivar (C)				
G	22.11 ± 0.42 ^a^	16.03 ± 0.37 ^a^	1.71 × 10^8^ ± 5.87 × 10^6^ ^a^	5048 ± 35 ^a^
R	20.72 ± 0.38 ^b^	15.92 ± 0.21 ^a^	1.82 × 10^8^ ± 5.77 × 10^6^ ^a^	3799 ± 27 ^b^
Interaction				
LVG	20.31 ± 0.43 ^b^	16.15 ± 0.56 ^a^	1.94 × 10^8^ ± 6.69 × 10^6^ ^a^	6753 ± 3 ^a^
LVR	19.76 ± 0.40 ^c^	15.73 ± 0.24 ^a^	1.92 × 10^8^ ± 6.30 × 10^6^ ^a^	5113 ± 40 ^b^
HVG	22.06 ± 0.41 ^a^	15.91 ± 0.18 ^a^	1.71 × 10^8^ ± 5.06 × 10^6^ ^b^	3343 ± 17 ^c^
HVR	21.27 ± 0.36 ^a^	15.91 ± 0.17a	1.71 × 10^8^± 5.25 × 10^6^ ^b^	2484 ± 15 ^c^
Significance				
VPD	***	NS	***	***
C	*	NS	NS	*
VPD × C	*	NS	NS	*

NS, *, ** and *** Not significant or significant at *p* < 0.05, 0.01 and 0.001, respectively.

**Table 3 plants-11-02369-t003:** Stomatal traits in terms of stomatal density (SD) and stomatal length (SL) from green (G) and red (R) Salanova plants grown under low (LV) and high (HV) VPD with differences within the leaf parts: b, bottom; m, medium; a, apex. All data are reported as mean ± se. Different letters indicate significant differences, according to Tukey test (*p* < 0.05).

	SD			SL		
		(n mm^2^)			(µm)	
VPD (V)						
LV	76.1 ± 2.43 ^a^			20.7 ± 0.35 ^b^		
HV	56.1 ± 2.71 ^b^			22.1 ± 0.33 ^a^		
Cultivar (C)						
G	64.9 ± 2.38 ^a^			21.4 ± 0.30 ^a^		
R	66.4 ± 2.75 ^b^			21.4 ± 0.37 ^a^		
Position (P)						
apex	95.0 ± 3.10 ^a^			22.0 ± 0.32 ^a^		
bottom	56.1 ± 2.48 ^b^			21.7 ± 0.34 ^b^		
medium	45.6 ± 2.12 ^c^			20.5 ± 0.35 ^c^		
Interaction (V × C × P)	b	m	a	b	m	a
LVG	50.8 ± 1.75 ^e^	62.1 ± 1.76 ^c^	110.6 ± 3.78 ^a^	20.2 ± 0.30 ^c^	20.5 ± 0.40 ^c^	18.8 ± 0.12 ^d^
LVR	49.0 ± 2.11 ^e^	62.4 ± 2.09 ^c^	111.9 ± 3.09 ^a^	20.8 ± 0.45 ^c^	20.0 ± 0.28 ^c^	17.4 ± 0.52 ^d^
HVG	39.0 ± 2.90 ^f^	51.7 ± 2.89 ^d^	78.1 ± 1.23 ^b^	28.4 ± 0.32 ^a^	23.5 ± 0.33 ^b^	24.1 ± 0.32 ^b^
HVR	39.4 ± 1.73 ^f^	52.4 ± 3.19 ^de^	79.4 ± 4.32 ^b^	27.4 ± 0.32 ^a^	23.5 ± 0.35 ^b^	23.9 ± 0.32 ^b^
Significance						
VPD		***			***	
C		NS			NS	
P		***			***	
V × C × P		NS			*	

NS, * and *** Not significant or significant at *p* < 0.05, 0.01 and 0.001, respectively.

**Table 4 plants-11-02369-t004:** Vein traits in terms of free vein endings (FEV) and vein density (VLA) from green (G) and red (R) Salanova plants grown under low (LV) and high (HV) VPD with differences within the leaf parts: b, bottom; m, medium; a, apex. All data are reported as mean ± se. Different letters indicate significant differences, according to Tukey test (*p* < 0.05).

	FEV			VLA		
		(n mm^−2^)			(mm mm^−2^)	
VPD (V)						
LV	18.8 ± 1.07 ^a^			3.64 ± 0.11 ^a^		
HV	10.2 ± 0.58 ^b^			2.67 ± 0.21 ^b^		
Cultivar (C)						
G	16.0 ± 0.84 ^a^			3.31 ± 0.24 ^a^		
R	16.8 ± 0.81 ^a^			3.13 ± 0.09 ^b^		
Position (P)						
apex	9.77 ± 0.95 ^c^			2.84 ± 0.11 ^c^		
bottom	12.5 ± 0.73 ^b^			3.19 ± 0.07 ^b^		
medium	19.2 ± 0.79 ^a^			3.63 ± 0.30 ^a^		
Interaction (V × C × P)	b	m	a	b	m	a
LVG	12.9 ± 0.71 ^c^	21.8 ± 0.98 ^b^	30.6 ± 0.54 ^a^	4.64 ± 0.13 ^a^	3.77 ± 0.11 ^b^	2.91 ± 0.15 ^c^
LVR	13.0 ± 0.93 ^c^	21.8 ± 0.92 ^b^	31.5 ± 0.32 ^a^	4.01± 0.14 ^a^	3.38 ± 0.06 ^b^	3.13 ± 0.06 ^c^
HVG	8.06 ± 0.87 ^d^	10.3 ± 0.55 ^cd^	12.6 ± 0.36 ^c^	3.06 ± 0.08 ^c^	2.85 ± 0.06 ^cd^	2.65 ± 0.88 ^d^
HVR	8.10 ± 0.66 ^d^	10.0 ± 0.46 ^cd^	12.0 ± 0.56 ^c^	2.92 ± 0.07 ^c^	2.76 ± 0.06 ^cd^	2.71 ± 0.12 ^d^
Significance						
VPD	***			***		
C	**			**		
P	***			***		
V × C × P	*			**		

NS, *, ** and *** Not significant or significant at *p* < 0.05, 0.01 and 0.001, respectively.

**Table 5 plants-11-02369-t005:** A list of measured traits and their measurement unit.

Trait	Acronym	Measurement Unit
Leaf size	LS	mm^2^
Leaf mass per area	LMA	g mm^−2^
Leaf dry matter content	LDMC	g g^−1^
Relative water content	RWC	%
Equivalent water thickness	EWT	g mm^−2^
Stomatal density	SD	n mm^−2^
Stomatal density per leaf size	SD × LS	n mm^−2^ LS
Stomatal guard cells length	SL	µm
Stomatal guard cells width	SW	µm
Epidermal cell density	ED	n mm^−2^
Epidermal cell density per leaf size	ED × LS	n mm^−2^ LS
Vein density	VLA	mm mm^−2^
Vein density per leaf size	VLA × LS	mm mm^−2^ LS
Free Vein Endings	FEV	n mm^−2^

## Data Availability

The data that support the findings of this study are available from the corresponding authors, upon reasonable request.

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
