# Peer review of "Vapour Pressure Deficit (VPD) Drives the Balance of Hydraulic-Related Anatomical Traits in Lettuce Leaves"

_plants, 2022, doi:10.3390/plants11182369_

Round 1

Reviewer 1 Report

This manuscript addresses the VPD effects on leaf morphological traits such as stomatal density and vein density associated with leaf size. My overall feeling is that the manuscript is well written and for the most part easy to follow, but could potentially be strengthened.

Line 68-74: To generalize the responses of leaf anatomy to high- and low-light growth conditions, the background evidence and information are poor. Please cite more articles related to sun and shade leaves. Also, this is quite an important part of this manuscript in order to emphasize the results of analyzing leaf anatomy in different positions of a leaf. Please strengthen the logical flow to address the key point answering “which plays a predominant role for leaf morphological traits such as stomatal density and vein density between VPD and irradiance?”.

Line 101: Lack of “.”

Line107-: Lots of abbreviations started to be appeared from a results section, which was not easy to follow for me. Please spell out full words when first mentioned in the text, although the journal rule might be different.

Line 116: Please describe why you measured EWT with its importance in leaf hydraulics.

Line 237: Sackand –> Sack and   

Line 293: stomata –> stomatal

Figure 5 legend: For me, green seems to be gray. The coloring of all figures is quite pale and not easy to see.

Line 413: Please rephrase the sentence “balance of leaf traits” because it is too ambiguous.

Table1: Why the RWC is so low? Is this unit of percentage?

Author Response

Reviewer 1

C1: This manuscript addresses the VPD effects on leaf morphological traits such as stomatal density and vein density associated with leaf size. My overall feeling is that the manuscript is well written and for the most part easy to follow, but could potentially be strengthened.

R1: Dear Reviewer 1, we would like to thank you for your invaluable revision, which has helped improve the overall quality of our manuscript.

C2: Line 68-74: To generalize the responses of leaf anatomy to high- and low-light growth conditions, the background evidence and information are poor. Please cite more articles related to sun and shade leaves. Also, this is quite an important part of this manuscript in order to emphasize the results of analyzing leaf anatomy in different positions of a leaf. Please strengthen the logical flow to address the key point answering “which plays a predominant role for leaf morphological traits such as stomatal density and vein density between VPD and irradiance?”.

R2: We added some more information in lines 83-100 to strengthen our point and justify our investigation.

C3: Line 101: Lack of “.”

R3: Done.

C4: Line107-: Lots of abbreviations started to be appeared from a results section, which was not easy to follow for me. Please spell out full words when first mentioned in the text, although the journal rule might be different.

R4: We added these specification in all the result section.

C5: Line 116: Please describe why you measured EWT with its importance in leaf hydraulics.

R5: We added some specification in lines 420-423.

C6:Line 237: Sackand –> Sack and   

R6: Done.

C7:Line 293: stomata –> stomatal

R7: Done.

C8:Figure 5 legend: For me, green seems to be gray. The coloring of all figures is quite pale and not easy to see.

R8: indeed the color it’s grey. We corrected the caption of Figure 5 now Figure S2 (following reviewer’s 2 suggestions). Colors were chosen using color-blind-friendly palettes according to the rules of the journal. We replaced green with grey in all legends.

C9:Line 413: Please rephrase the sentence “balance of leaf traits” because it is too ambiguous.

R9: Done. Line 492.

C10:Table1: Why the RWC is so low? Is this unit of percentage?

R10: We thank the reviewer 2 for noticing this flaw and corrected the mistake in table1 and reported the corrected values. RWC is expressed in %.

Reviewer 2 Report

The authors investigated the development of leaf anatomical traits in lettuce plants growing under different microclimate, focusing on low and high VPD level. Leaf anatomy was strongly influenced by the VPD treatment and furthermore also different along the leaf, showing a gradient in microclimatic conditions, e.g. light level and VPD. The experimental setup as well as design are without flaw and the results are very interesting. However, several points should be addressed before publication:

-The hypothesis should be formulated clearer and later also be answered in the discussion

-Statistical analysis should be improved (see detailed comments)

-Data presentation is sometimes a little confuse and the order of figures should be reconsidered

-The results are often repeated in the discussion, which should be reduced

-Some statements in the discussion are very assumptive and should be extended, e.g. by including the role of CO2 uptake in some points. If the authors have any data regarding the water use efficiency or chlorophyll content along the leaf, it would be great to add them here. Such information might ease the discussion and make it less speculative.

However, all together the paper is nicely written and for sure of high interest for future greenhouse studies or crop production in controlled environments. For detailed comments please check the attached PDF file.

Author Response

C11: The authors investigated the development of leaf anatomical traits in lettuce plants growing under different microclimate, focusing on low and high VPD level. Leaf anatomy was strongly influenced by the VPD treatment and furthermore also different along the leaf, showing a gradient in microclimatic conditions, e.g. light level and VPD. The experimental setup as well as design are without flaw and the results are very interesting. However, several points should be addressed before publication:

-The hypothesis should be formulated clearer and later also be answered in the discussion

-Statistical analysis should be improved (see detailed comments)

-Data presentation is sometimes a little confuse and the order of figures should be reconsidered

-The results are often repeated in the discussion, which should be reduced

-Some statements in the discussion are very assumptive and should be extended, e.g. by including the role of CO2 uptake in some points. If the authors have any data regarding the water use efficiency or chlorophyll content along the leaf, it would be great to add them here. Such information might ease the discussion and make it less speculative.

However, all together the paper is nicely written and for sure of high interest for future greenhouse studies or crop production in controlled environments. For detailed comments please check the attached PDF file.

R11: Dear reviewer 2, we are very pleased that you have acknowledged the interest of our results. Following your comments, we have improved the paper and we hope it is now suitable for publication.

C12: The abstract contains a lot of introduction but hardly gives any details on measured parameters or methods. Also the results mentioned here are very vague and the disscusion/conclusion part is almost completely missing.
R12: We modified the whole abstract following reviewer’s 2 suggestions.

C13: Line 9: Not only, also physiology plays an important role here...
R13: Done. Line 10.

C14: Line 14: What traits?
R14: We clarified the concept. Line 14.  

C15: Lines 29-31: I think plant physiology has always been of high interest...
R15: We corrected the sentence. Line 34.

C16: Lines 34-35: how?
R16: It is explained in the following lines. However, we modified the sentence for clarity. Line 41.

C17: Line 37: Do you have any citations for this? Because i cant find any data about this in Wang et al 2020. Otherwise i suggest its simply a lack of water and potentially hydraulic failure rather than overheating...
R17: We better clarified the state reported in Wang et al. 2020. Lines 43-45.

C18: Lines 45-47: That might be possible for plants growing in an controlled environment but in nature mostly leaves are developed under "low" VPD in spring and only later on in summer subjected to high VPD.
R18: In this lines we are not talking about a controlled environment but hostile environments, where for example there is little water available, as reported in lines 53-55.

C19: Lines 48-50: Isnt this very species specific and also dependent on the photosynthetic pathway (C3 vs C4 ca CAM)
R19: We agree that it depends on C3, C4 or CAM metabolism. We added a comment on the dependence on species-specific mechanisms without going too in-depth because it was out of the scope of the paper. Lines 58-60.

C20: Line 55: Decreasing?
R20: “Diluted” is the technical correct term here(see also the reference for clarification).

C21: Lines 56-58: Only if leaf development happens already under high VPD and low water availability. Which rarely happens in nature as mostly spring is moister than late summer.
R21: We specified that we are talking about sub-optimal/harsh environments. Line 67.

C22: Lines 68-70: I think there are many more interesting points here: light compensation point, photorespiration, maximum assimilation rate, etc
R22: Yes, we added some more information on this in lines 83-100.

C23: Lines 78-80: Citation missing
R23: Done. Line 107.

C24: Lines 92-94: Why?
R24: We added some clarification in lines 117,121.

C25: Lines 95-100: To me hypothesis should be questions which can be rejected or accepted. These objectives here sound rather vague and generic, especially number 3.
R25:  We modified this part following the suggestion. Lines 124-128.

C26: Since the MM part is not shown here before the Results all abbreviations need to be explained here!
R26: Done.

C27: Lines 111-115: p-values missing and the differences are between cultivars, treatments or the interaction?
R27:P-value are specified in all the tables under the voice “significance” as explained by the subscripts and in this case in lines 138-140.

C28: Figure 1 caption: This data is not only driven by stomatal density but also by leaf size. Therefore its very hard to disentangle differences. I suggest to rather show LS and SD separatly. This could be a figure for appendix.
R28: These data are reported in supplemental material and leaf size is reported in table 1.

C29: Figure 3 caption: Of course these correlate becuase in both parameters the leaf area is already included. So why not use VLA and SD?
R29: They are showed in Figure 4.

C30: Lines 167-168: Why bold?
R30: We corrected the mistake.

C31: Line 171: This section is very hard to read with all the abbreviations...
R31: We modified this paragraph to make it easier to read.

C32: Table 3 caption: Why is this shown so late in the results part? I would show this data before showing any combined data (SD x LS)
R32: Because in this table, the results of the effects of third factor (the position of leaf region analyzed, thus the three-way anova) are also reported to respond to the second question/hypothesis of the study.

C33: Figure 4 caption : Why is this split up in 6 sub plots and not only 3?
R33: This was meant to avoid overlaps and make the graphs clearer.

C34: Figure 5 caption: What does this graph show? Is there any relationship in it? And isnt this very much the same as Fig 4?
R34: We agree and moved this figure to supplemental. Now Figure S2.

C35: Line 214 : Citations missing
R35: Done. Line 279.

C36: Line 221: Thats a pretty strong statement which was not really tested here.
R36: We smoothed the statement. Line 285.

C37: Line 227- 231 : This was already mentioned in the introduction
R37: We removed the sentence to avoid repetition.

C38: Lines 241-243: If this is already clear, why did you test it?
R38: We modified the sentence to avoid misunderstanding. As stated in the introduction section, results are often unclear and little is known in crops and especially in lettuce species.

C39: Line 269: Drought stress (too little water) or water stress (too much water)?
R39: Water availability sounds better here. Line 334. 

C40: Lines 287-289: In L 241-242 you stated that the number of stomata and veins must correlate. How does this fit here?
R40: We added a clarification. Please also refer to R38.

C41: Lines 298-302: The situation in a forest can hardly be compared with a lettuce. I strongly suggest to remove this part.
R41: We removed the sentence to avoid confusion.

C42: Line 316: I dont really see how this can be applied in nature. In nature the climate is highly fluctuating and rarely stable between days. additionally in nature plants mostly flush under low VPD and lower radiation while only latter in the year higher VPDs occur.
R42: This is referred to the sudden changes in the climate happening today in semi-arid and arid regions. We added a few words to clarify. Line 383-384.

C43: Line 327: Thats a rather low light level. Is this usually used in greenhouses to produce crops?
R43: We are talking about indoor facilities (growth chamber and vertical farming) and, actually in indoor production facility this is a usual light level.

C44: Line 328: So your n = 9 for each treament?
R44: 9 green and 9 red, 18 for VPD condition. We specified it in line 395.

C45: Line 329-330: What are the reasons for these values, e.g. 24°C? Is this the usual temperature in greenhouses or what are standard settings in commercial greenhouses?
R45: Yes. Please refer also to R43.

C46: Table 5: To me this is the absolute number of stomata in one leaf. So its not a density! This must be clarified and if possible renamed! Same goes for ED x Ls and VLA x LS
R46: We agree and amended the measurement unit to make it clearer. We also added some clarifications in the material and methods section (Lines 426-429,437,444). Anyway we prefer to keep the name “VLA” and “SD” in order to make it clearer that we are still talking about the same traits calculated not only for a portion of the leaf but also for the entire lamina as also reported in other previous studies like Carins Murphy et al. 2014 (doi: 10.1111/pce.12136).

C47: Lines 350-351: What is this good for?
R47: We added some clarification. Lines 420-423.

C48: Line 355: By what rules? One third of the leaf for each section?
R48: yes, we added some clarification. Line 426.

C49: Line 363: Were these values then averaged per leaf to avoid pseudo-replication?
R49: All the values were averaged per condition.

C50: Line 382: When was this measured? When stomata were completely open? I think this is a very plastic parameter depending on many details.
R50: We added some clarification for all anatomical analyses sampling procedures. Lines 427-429.

C51: Lines 401-403: How did you correct for the fact that samples from the same leaf but different areas are not independent from each other? I suggest to use a mixed effect model and add a random effect as leaf identity.
R51: The “position” was used as a variability factor with the leaf as the “sample” and the portions of leaf in different positions as an independent “subsamples”. All the samplings were performed at the same time, as quickly as possible to standardize the condition. Please also refer to R50.

C52: Line 403: What about homogeneity of variances?
R52: Done. Line 481-482.

C53: Line 406: What kind? Spearman, Pearson, ...?
R53: Done. Line 488-489.

C54: Line 413: balance what exactly?
R54: We modified this sentence. Line 493.

C55: Line 417: What about your won hypothesis? Where do you answer them?
R55: We added some more information about it. Lines 493-499.

C56: Line 430: Conditions?
R56: to changes in the environment. We modified the sentence to make it clearer. Line 511.

Round 2

Reviewer 2 Report

The authors did a good job during and the manuscript should now be ready for publication.